# CONTRASTIVE SYN-TO-REAL GENERALIZATION

**Wuyang Chen**[1]*, **Zhiding Yu**[2]†, **Shalini De Mello**[2], **Sifei Liu**[2], **Jose M. Alvarez**[2],
**Zhangyang Wang**[1], **Anima Anandkumar**[2,3]
[1]The University of Texas at Austin   [2]NVIDIA   [3]California Institute of Technology
{wuyang.chen,atlaswang}@utexas.edu
{zhidingy,shalinig,sifeil,josea,aanandkumar}@nvidia.com
https://github.com/NVlabs/CSG

## ABSTRACT

Training on synthetic data can be beneficial for label or data-scarce scenarios. However, synthetically trained models often suffer from poor generalization in real domains due to domain gaps. In this work, we make a key observation that the diversity of the learned feature embeddings plays an important role in the generalization performance. To this end, we propose contrastive synthetic-to-real generalization (CSG), a novel framework that leverages the pre-trained ImageNet knowledge to prevent overfitting to the synthetic domain, while promoting the diversity of feature embeddings as an inductive bias to improve generalization. In addition, we enhance the proposed CSG framework with attentional pooling (A-pool) to let the model focus on semantically important regions and further improve its generalization. We demonstrate the effectiveness of CSG on various synthetic training tasks, exhibiting state-of-the-art performance on zero-shot domain generalization.

## 1 INTRODUCTION

Deep neural networks have pushed the boundaries of many visual recognition tasks. However, their success often hinges on the availability of both training data and labels. Obtaining data and labels can be difficult or expensive in many applications such as semantic segmentation, correspondence, 3D reconstruction, pose estimation, and reinforcement learning. In these cases, learning with synthetic data can greatly benefit the applications since large amounts of data and labels are available at relatively low costs. For this reason, synthetic training has recently gained significant attention (Wu et al., 2015; Richter et al., 2016; Shrivastava et al., 2017; Savva et al., 2019).

Despite many benefits, synthetically trained models often have poor generalization on the real domain due to large domain gaps between synthetic and real images. Limitations on simulation and rendering can lead to degraded synthesis quality, such as aliased boundaries, unrealistic textures, fake appearance, over-simplified lighting conditions, and unreasonable scene layouts. These issues result in domain gaps between synthetic and real images, preventing the synthetically trained models from capturing meaningful representations and limiting their generalization ability on real images.

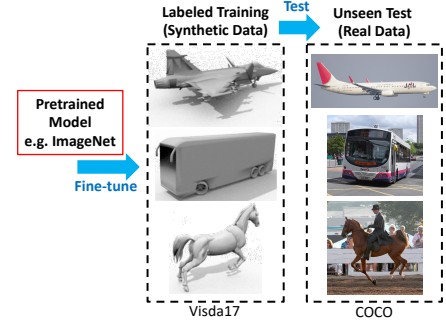

**Figure 1:** An illustration of the domain generalization protocol on the VisDA-17 dataset, where real target domain (test) images are assumed unavailable during model training.

To mitigate these issues, domain generalization and adaptation techniques have been proposed (Li et al., 2017; Pan et al., 2018; Yue et al., 2019). Domain adaptation assumes the availability of target data (labeled, partially labeled, or unlabeled) during training. On the other hand, domain generalization considers zero-shot generalization without seeing the target data of real images, and is therefore more challenging. An illustration of the domain generalization protocol on the

---

*Work done during the research internship with NVIDIA.
†Corresponding author.

VisDA-17 dataset (Peng et al., 2017) is shown in Figure 1. Considering that ImageNet pre-trained representation is widely used as model initialization, recent efforts on domain generalization show that such knowledge can be used to prevent overfitting to the synthetic domain (Chen et al., 2018; 2020c). Specifically, they impose a distillation loss to regularize the distance between the synthetically trained and the ImageNet pre-trained representations, which improves synthetic-to-real generalization.

The above approaches still face limitations due to the challenging nature of this problem. Taking a closer look, we observe the following pitfalls in training on synthetic data. First, obtaining photo-realistic appearance features at the micro-level, such as texture and illumination, is challenging due to the limits of simulation complexity and rendering granularity. Without special treatment, CNNs tend to be biased towards textures (Geirhos et al., 2019) and suffer from badly learned representations on synthetic data. Second, the common lack of texture and shape variations on synthetic images often leads to collapsed and trivial representations without any diversity. This is unlike training with natural images where models get sufficiently trained by seeing enough variations. Such a lack of diversity in the representation makes the learned models vulnerable to natural variations in the real world.

**Summary of contributions and results:**

- We observe that the diversity of learned feature embedding plays an important role in synthetic-to-real generalization. We show an example of collapsed representations learned by a synthetic model, which is in sharp contrast to features learned from real data (Section 2).

- Motivated by the above observation, we propose a contrastive synthetic-to-real generalization framework that simultaneously regularizes the synthetically trained representation while promoting the diversity of the learned representation to improve generalization (Section 3.1).

- We further enhance the CSG framework with attentional pooling (A-pool) where feature representations are guided by model attention. This allows the model to localize its attention to semantically more important regions, and thus improves synthetic-to-real generalization (Section 3.4).

- We benchmark CSG on various synthetic training tasks including image classification (VisDA-17) and semantic segmentation (GTA5 → Cityscapes). We show that CSG considerably improves the generalization performance without seeing target data. Our best model reaches 64.05% accuracy on VisDA-17 compared to previous state-of-the-art (Chen et al., 2020c) with 61.1% (Section 4).

## 2 A MOTIVATING EXAMPLE

We give a motivating example to show the significant differences between the features learned on synthetic and real images. Specifically, we use a ResNet-101 backbone and extract the $l_2$ normalized feature embedding after global average pooling (defined as $\bar{v}$). We consider the following three models: 1) model pre-trained on ImageNet, 2) model trained on VisDA-17 validation set (real images), and 3) model trained on VisDA-17 training set (synthetic images) [1]. Both 2) and 3) are initialized with ImageNet pre-training, and fine-tuned on the 12 classes defined in VisDA-17.

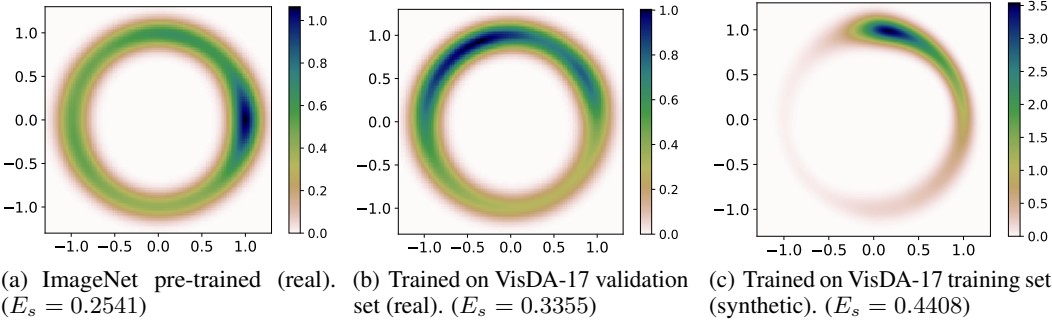

(a) ImageNet pre-trained (real). ($E_s = 0.2541$)

(b) Trained on VisDA-17 validation set (real). ($E_s = 0.3355$)

(c) Trained on VisDA-17 training set (synthetic). ($E_s = 0.4408$)

**Figure 2:** Feature diversity on VisDA-17 test images in $\mathbb{R}^2$ with Gaussian kernel density estimation (KDE). Darker areas have more concentrated features. $E_s$: hypersphorical energy of features, lower the more diverse.

---

[1] For a fair comparison, we make a random subset of the training set with an equal size of the validation set, since the training set of VisDA-17 is larger than the validation set.

**Visualization of feature diversity.** We visualize the normalized representations on a 2-dim sphere. A Gaussian kernel with bandwidth estimated by Scott's Rule (Scott, 2015) is applied to estimate the probability density function. Darker areas have more concentrated features, and if the feature space (the 2-dim sphere) is covered by dark areas, it has more diversely placed features. In Figure 2, we can see that the ImageNet pretrained model can widely span the representations on the 2-dim feature space. The model trained on VisDA-17 validation set can also generate diverse features, although slightly affected by the class imbalance. However, when the model is trained on the training set (synthetic images), the features largely collapse to a narrow subspace, i.e., the model fails to fully leverage the whole feature space. This is clear that training on synthetic images can easily introduce poor bias to the model and the collapsed representations will fail to generalize to the real domain.

**Quantitive measurement of feature diversity.** Inspired by (Liu et al., 2018), we also quantitatively measure the diversity of the feature embeddings using the following hyperspherical potential energy:

$$E_s\left(\bar{\boldsymbol{v}}_i|_{i=1}^N\right) = \sum_{i=1}^N \sum_{j=1, j\neq i}^N e_s\left(\|\bar{\boldsymbol{v}}_i - \bar{\boldsymbol{v}}_j\|\right) = \begin{cases} \sum_{i\neq j}\|\bar{\boldsymbol{v}}_i - \bar{\boldsymbol{v}}_j\|^{-s}, & s > 0 \\ \sum_{i\neq j}\log\left(\|\bar{\boldsymbol{v}}_i - \bar{\boldsymbol{v}}_j\|^{-1}\right), & s = 0 \end{cases} \quad (1)$$

$N$ is the number of examples. The lower the hyperspherical energy (HSE) is, the more diverse the feature vectors will be scattered in the unit sphere. $s$ is the power factor, and we choose $s = 0$ in this example. Three training strategies exhibit energies as $0.2541, 0.3355, 0.4408$, respectively. This validates that models trained on real images can capture diverse features, whereas the synthetic training will lead the model to highly collapsed feature space.

**Remarks.** A conclusion can be drawn from the above examples: though assisted with ImageNet initialization, fine-tuning on synthetic images tends to give collapsed features with poor diversity in sharp contrast to training with real images. This indicates that the diversity of learned representation could play an important role in synthetic-to-real generalization.

## 3    CONTRASTIVE SYNTHETIC-TO-REAL GENERALIZATION

We consider the synthetic-to-real domain generalization problem following the protocols of Chen et al. (2020c). More specifically, the objective is to achieve the best zero-shot generalization on the unseen target domain real images without having access to them during synthetic training.

### 3.1    NOTATION AND FRAMEWORK

Our design of the model considers the following two aspects with a "push and pull" strategy:
**Pull:** Without access to real images, the ImageNet pre-trained model presents the only source of real domain knowledge that can implicitly guide our training. As a result, we hope to impose some form of similarity between the features obtained by the synthetic model and the ImageNet pre-trained one. This helps to overcome the domain gaps from the unrealistic appearance of synthetic images.
**Push:** Section 2 shows that synthetic training tends to generate collapsed features whereas models trained on natural images give many diverse ones. We treat this as an inductive bias to improve synthetic training, by pushing the feature embeddings away from each other across different images.

The above "push and pull" strategy can be exactly formulated with a contrastive loss. This motivates us to propose a contrastive synthetic-to-real generalization framework as partly inspired by recent popular contrastive learning methods (He et al., 2020). Figure 3(b) illustrates our CSG framework. Specifically, we denote the frozen Imagenet pre-trained model as $f_{e,o}$ and the synthetically trained model $f_e$, where $f_e$ is supervised by the task loss $\mathcal{L}_{syn}$ for the defined downstream task. We denote the input synthetic image as $\boldsymbol{x}^a$ and treat it as an anchor. We treat the embeddings of $\boldsymbol{x}^a$ obtained by $f_e$ and $f_{e,o}$ as anchor and positive embeddings, denoting them as $\boldsymbol{z}^a$ and $\boldsymbol{z}^+$, respectively. Following a typical contrastive approach, we define $K$ negative images $\{\boldsymbol{x}_1^-, \cdots, \boldsymbol{x}_K^-\}$ for every anchor $\boldsymbol{x}^a$, and denote their corresponding embeddings as $\{\boldsymbol{z}_1^-, \cdots, \boldsymbol{z}_K^-\}$. Similar to the design in (Chen et al., 2020d), we define $h/\widetilde{h} : \mathbb{R}^C \to \mathbb{R}^c$ as the nonlinear projection heads with a two MLP layers and a ReLU layer between them. The CSG framework regularizes $f_e$ in a contrastive manner: pulling $\boldsymbol{z}^a$ and $\boldsymbol{z}^+$ to be closer while pushing $\boldsymbol{z}^a$ and $\{\boldsymbol{z}_1^-, \cdots, \boldsymbol{z}_K^-\}$ apart. This regularizes the model by preventing its representation from deviating too far from that of a pre-trained ImageNet model and yet encouraging it to learn task-specific information from the synthetic data.

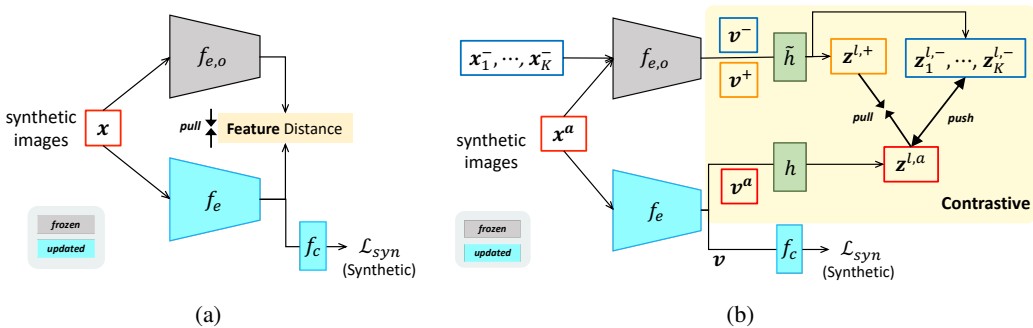

**Figure 3:** (a) Previous work (Chen et al., 2018; 2020c) consider "learning without forgetting" which minimizes a distillation loss between a synthetic model and an ImageNet pre-trained one (either on features or model parameters) to avoid catastrophic forgetting. (b) The proposed CSG framework with a "push and pull" strategy.

Even though having connections to recent self-supervised contrastive representation learning methods (Oord et al., 2018; Wu et al., 2018; Chen et al., 2020a; He et al., 2020; Chen et al., 2020b; Jiang et al., 2020), our work differs in the following aspects: 1) Self-supervised learning and the addressed task are ill-posed in different manners - the former lacks the constraints from semantic labels, whereas the latter lacks the support of data distribution. 2) As a result, the motivations of contrastive learning are different. Our work is also related to the contrastive distillation framework in (Tian et al., 2020a). Again, the two works differ in both task and motivation despite the converging techniques.

### 3.2 AUGMENTATION

Augmentation has been an important part of effective contrastive learning. By perturbing or providing different views of the representations, augmentation forces a model to focus more on the mid-level and high-level representations of object parts and structures which are visually more realistic and reliable. To this end, we follow existing popular approaches to create augmentation at different levels:

**Image augmentation.** We consider image-level augmentation using RandAugment (Cubuk et al., 2020) where a single global control factor $M$ is used to control the augmentation magnitude. We denote the transform operators of image-level augmentation as $\mathcal{T}(\cdot)$.

**Model augmentation.** We adopt a mean-teacher (Tarvainen & Valpola, 2017) styled moving average of a model to create different views of feature embeddings. Given an anchor image $x^a$ and $K$ negative images $\{x_1^-, \cdots, x_K^-\}$, we compute the embeddings as follows:

$$z^a = f_e \circ g \circ h(\mathcal{T}(x^a)), \ z^+ = f_{e,o} \circ g \circ \widetilde{h}(\mathcal{T}(x^a)), \ z_k^- = f_{e,o} \circ g \circ \widetilde{h}(\mathcal{T}(x_k^-)), \quad (2)$$

where $g : \mathbb{R}^{C \times h \times w} \to \mathbb{R}^C$ is a pooling operator transforming a feature map into a vector. Following (He et al., 2020), we define $\widetilde{h}(\cdot)$ as an exponential moving average of the $h(\cdot)$ across different iterations. Such difference in $h(\cdot)$ and $\widetilde{h}(\cdot)$ leads to augmented views of embeddings.

### 3.3 CONTRASTIVE LOSS

We use InfoNCE loss (Wu et al., 2018) to formulate the "push and pull" strategy:

$$\mathcal{L}_{\text{NCE}} = -\log \frac{\exp\left(z^a \cdot z^+/\tau\right)}{\exp\left(z^a \cdot z^+/\tau\right) + \sum_{z^-} \exp\left(z^a \cdot z^-/\tau\right)}, \quad (3)$$

where $\tau = 0.07$ is a temperature hyper-parameter in our work. Together, we minimize the combination of the synthetic task loss and $\mathcal{L}_{\text{NCE}}$ during our transfer learning process:

$$\mathcal{L} = \mathcal{L}_{\text{Task}} + \lambda \mathcal{L}_{\text{NCE}} \quad (4)$$

Specifically, $\mathcal{L}_{\text{Task}}$ is the synthetic training task objective. For example, $\mathcal{L}_{\text{Task}}$ is a cross-entropy loss of a vector over the 12 defined classes on VisDA-17, whereas it is a per-pixel dense cross-entropy loss on GTA5. $\lambda$ is a balancing factor controlling the strength of the Contrastive Learning.

**Multi-layer contrastive learning.** We are curious that on which layer(s) should we apply contrastive learning to achieve best generalization. We therefore propose a multi-layer CSG framework with different groups (combinations) of layer, denoted as $\mathcal{G}$:

$$\mathcal{L}_{\text{NCE}} = \sum_{l \in \mathcal{G}} \mathcal{L}_{\text{NCE}}^l = \sum_{l \in \mathcal{G}} - \log \frac{\exp\left(\boldsymbol{z}^{l,a} \cdot \boldsymbol{z}^{l,+}/\tau\right)}{\exp\left(\boldsymbol{z}^{l,a} \cdot \boldsymbol{z}^{l,+}/\tau\right) + \sum_{\boldsymbol{z}^{l,-}} \exp\left(\boldsymbol{z}^{l,a} \cdot \boldsymbol{z}^{l,-}/\tau\right)} \tag{5}$$

We conduct an ablation in Section 4.1.2 to study the generalization performance with respect to different $\mathcal{G}$ on ResNet-101[2]. Note that the non-linear projection heads $h^l(\cdot)/\widetilde{h}^l(\cdot)$ are layer-specific.

**Cross-task dense contrastive learning.** Semantic segmentation presents a new form of task with per-pixel dense prediction, and the task naturally requires pixel-wise dense supervision $\mathcal{L}_{\text{Task}}$. Unlike image classification, an image in semantic segmentation could contain rich amounts of objects. We therefore make $\mathcal{L}_{\text{NCE}}$ spatially denser in semantic segmentation to make it more compatible with the dense task loss $\mathcal{L}_{\text{Task}}$. Specifically, the NCE losses are applied on cropped feature map patches:

$$\mathcal{L}_{\text{NCE}} = \sum_{l \in \mathcal{G}} \sum_{i=1}^{N_l} \mathcal{L}_{\text{NCE}}^{l,i} = \sum_{l \in \mathcal{G}} \sum_{i=1}^{N_l} -\frac{1}{N_l} \log \frac{\exp\left(\boldsymbol{z}_i^{l,a} \cdot \boldsymbol{z}_i^{l,+}/\tau\right)}{\exp\left(\boldsymbol{z}_i^{l,a} \cdot \boldsymbol{z}_i^{l,+}/\tau\right) + \sum_{\boldsymbol{z}_i^{l,-}} \exp\left(\boldsymbol{z}_i^{l,a} \cdot \boldsymbol{z}_i^{l,-}/\tau\right)} \tag{6}$$

where we crop $\boldsymbol{x}^a$ into local patches $\boldsymbol{x}_i^a$ with $\boldsymbol{z}_i^a = f_e \circ g \circ h(\mathcal{T}(\boldsymbol{x}_i^a))$. Similar for $\boldsymbol{x}^-$. In practice, we crop $\boldsymbol{x}$ into $N_l = 8 \times 8 = 64$ local patches during segmentation training.

### 3.4 A-POOL: ATTENTIONAL POOLING FOR IMPROVED REPRESENTATION

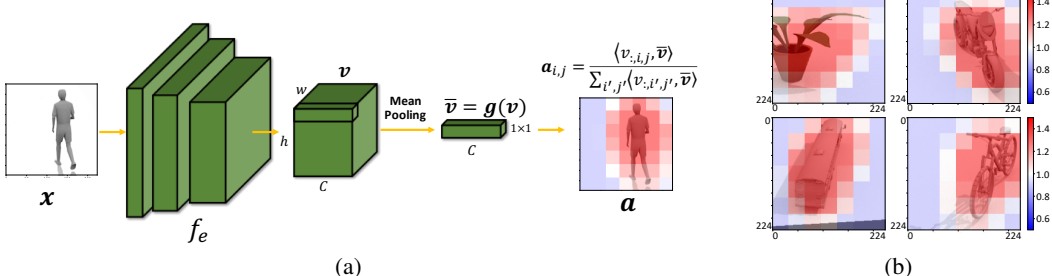

(a)

(b)

**Figure 4:** (a) For each input image, A-pool computes an attention matrix $\boldsymbol{a}$ based on the inner product between the global average pooled feature vector $\overline{\boldsymbol{v}}$ and vector at each position $\boldsymbol{v}_{:,i,j}$ ($\overline{\boldsymbol{v}}, \boldsymbol{v}_{:,i,j} \in \mathbb{R}^C$). (b) Example of four generated reweighting matrices on different images. Note that the values are defined as the ratio of the attention over uniform weight. The attention is visualized with upsampling to match the input size (224×224).

The purpose of the pooling function $g(\cdot)$ and the non-linear projection head $h(\cdot)$ is to project a high dimensional feature map $\boldsymbol{v}$ from $\mathbb{R}^{C \times h \times w}$ to a low-dimensional embedding in $\mathbb{R}^c$. With the feature pooled by $g(\cdot)$ being more informative, we could also let the contrastive learning focus on more semantically meaningful representations. Inspired by recent works showing CNN's capability of localizing salient objects (Zhou et al., 2016; Zhang et al., 2018) with only image-level supervision, we propose an attentional pooling (A-pool) module to improve the quality of the pooled feature.

As shown in Figure 4(a), given a feature map $\boldsymbol{v}$ we first calculate its global average pooled vector $\overline{\boldsymbol{v}} = g(\boldsymbol{v}) = \frac{1}{hw}[\sum_{i,j} \boldsymbol{v}_{1,i,j}, \cdots, \sum_{i,j} \boldsymbol{v}_{C,i,j}], i \in [1, h], j \in [1, w]$, we then define the attention score for each pixel at $(i, j)$ as $\boldsymbol{a}_{i,j} = \frac{\langle \boldsymbol{v}_{:,i,j}, \overline{\boldsymbol{v}} \rangle}{\sum_{i',j'} \langle \boldsymbol{v}_{:,i',j'}, \overline{\boldsymbol{v}} \rangle}$ ($i' \in [1, h], j' \in [1, w]$) and use this score as the weight term in global pooling. Specifically, we define A-pool operator as $\hat{\boldsymbol{v}} = g_a(\boldsymbol{v}) = [\sum_{i,j} \boldsymbol{v}_{1,i,j} \cdot \boldsymbol{a}_{i,j}, \cdots, \sum_{i,j} \boldsymbol{v}_{C,i,j} \cdot \boldsymbol{a}_{i,j}]$. This attention-weighted pooling procedure can effectively shift the focus of the pooled feature vector to the semantically salient regions, leading to more meaningful contrastive learning. In Figure 4(b), we plot the attention as the ratio of new attention score $\boldsymbol{a}$ over uniform weights (i.e., the uniform score used in global average pooling as $\frac{1}{h \times w}$. For example, a value 1.5 in Figure 4(b) indicates an attention score of $\frac{1.5}{h \times w}$). Note that if any spatially-related augmentation is applied, the attention used for $f_{e,o}$ as described in section 3.4 will be calculated by $f_e$, since $f_e$ is the one adapted to the source domain with better attention.

---

[2]We follow the design by He et al. (2016) to group the convolution operators with the same input resolution as a layer which results in four layer groups in ResNet-101.

# 4 EXPERIMENT

We follow (Chen et al., 2020c) to evaluate on two popular benchmarks: VisDA-17→COCO (classification) and GTA5→Cityscapes (segmentation). Codes is available at `https://github.com/NVlabs/CSG`.

## 4.1 IMAGE CLASSIFICATION

**Dataset.** The VisDA-17 dataset (Peng et al., 2017) provides three subsets (domains), each with the same 12 object categories. Among them, the training set (source domain) is collected from synthetic renderings of 3D models under different angles and lighting conditions, whereas the validation set (target domain) contains real images cropped from the Microsoft COCO dataset (Lin et al., 2014).

**Implementation.** For VisDA-17, we choose ImageNet pretrained ResNet-101 (He et al., 2016) as the backbone. We fine-tune the model on the source domain with SGD optimizer of learning rate $1 \times 10^{-4}$, weight decay $5 \times 10^{-4}$, and momentum 0.9. Batch size is set to 32, and the model is trained for 30 epochs. $\lambda$ for $\mathcal{L}_{\text{NCE}}$ is set as 0.1.

### 4.1.1 MAIN RESULTS

We compare with different distillation strategies in Table 1, including feature $l_2$ regularization (Chen et al., 2018), parameter $l_2$ regularization, importance weighted parameter $l_2$ regularization (Zenke et al., 2017), and KL divergence (Chen et al., 2020c). All these approaches try to retain the ImageNet domain knowledge during the synthetic training, without feature diversity being explicitly promoted. One could see, CSG significantly improves generalizaiton performance over these baselines.

We also verify that CSG promotes diverse representations, and that the diversity is correlated with generalization performance. To this end, we quantitatively measure the hyperspherical energy defined in Equation 1 on the feature embeddings extracted by different methods. From Table 1, one can see that the baseline suffers from the highest energy, and under different power settings, CSG consistently achieves the lowest energies. Table 1 indicates that a method that achieves lower HSE can better generalize from synthetic to the real domain. This confirms our motivation that forcing the model to capture more diversely scattered features will achieve better generalization performance.

**Table 1:** Generalization performance and hyperspherical energy of the features extracted by different models (lower is better). Dataset: VisDA-17 (Peng et al., 2017) validation set. Model: ResNet-101.

| Model | Power | | | Accuracy (%) |
|---|---|---|---|---|
| | 0 | 1 | 2 | |
| Oracle on ImageNet[3] | - | - | - | 53.3 |
| Baseline (vanilla synthetic training) | 0.4245 | 1.2500 | 1.6028 | 49.3 |
| Weight $l2$ distance (Kirkpatrick et al., 2017) | 0.4014 | 1.2296 | 1.5302 | 56.4 |
| Synaptic Intelligence (Zenke et al., 2017) | 0.3958 | 1.2261 | 1.5216 | 57.6 |
| Feature $l2$ distance (Chen et al., 2018) | 0.3337 | 1.1910 | 1.4449 | 57.1 |
| ASG (Chen et al., 2020c) | 0.3251 | 1.1840 | 1.4229 | 61.1 |
| CSG (Ours) | **0.3188** | **1.1806** | **1.4177** | **64.05** |

### 4.1.2 ABLATION STUDY

We perform ablation studies (Table 2, 3, 4) on the VisDA-17 image classification benchmark (Peng et al., 2017).

**Augmentation.** We study different magnitudes of RandAugment (Cubuk et al., 2020) in our scenario (Section 3.2), as summarized in Table 2. By tuning the global magnitude control factor $M$, we observe that too strong augmentations deteriorate generalization (e.g. $M = 12, 18, 24$), while mild augmentation brings limited help ($M = 3$). A moderate augmentation ($M = 6$) can improve contrastive learning.

---

[3]The oracle is obtained by freezing the ResNet-101 backbone while only training the last new fully-connected classification layer on the VisDA-17 source domain (the FC layer for ImageNet remains unchanged). We use the PyTorch official model of ImageNet-pretrained ResNet-101.

**Multi-layer Contrastive Learning.** Since features from the high-level layers are directly responsible for the downstream classification or other vision tasks, we suspect the last layer in the feature extractor $f_e$ would be the most important. We conduct an ablation study on generalization performance with different layer combinations for multi-layer contrastive learning (Section 3.3). From Table 3, one can see that applying $\mathcal{L}_{\text{NCE}}$ on layer 3 and 4 are most effective. Therefore, in our work we set $\mathcal{G} = \{3, 4\}$

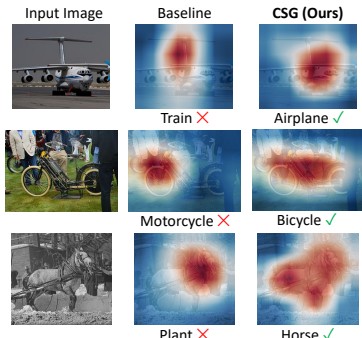

**Figure 5:** An illustration of model attention by GradCAM (Selvaraju et al., 2017) on the VisDA-17 validation set.

**A-pool.** Table 4 shows that with attention guided pooling (Section 3.4), A-pool can further improve the generalization performance, compared with the vanilla global average pooling (GAP).

**Table 2:** Ablation with $M$.

| $M$ | Accuracy |
|---|---|
| 0 (no aug.) | 60.86 |
| 3 | 61.36 |
| 6 | **62.88** |
| 12 | 62.61 |
| 18 | 62.00 |

**Table 3:** Ablation with $\mathcal{G}$.

| Layer Groups $\mathcal{G}$ | Accuracy (%) |
|---|---|
| 4 | 62.88 |
| 3+4 | **63.77** |
| 2+3+4 | 62.66 |
| 1+2+3+4 | 62.30 |

**Table 4:** Ablation w./w.o. A-pool. GAP: global average pooling.

| Pooling | Accuracy (%) |
|---|---|
| GAP | 63.77 |
| A-pool | **64.05** |

### 4.1.3 CSG BENEFITS VISUAL ATTENTION

We further show the Grad-CAM[3] attention on VisDA-17 validation set (Figure 5). We can see that our CSG framework also contributes to better visual attention on unseen real images.

## 4.2 SEMANTIC SEGMENTATION

**Dataset.** GTA5 (Richter et al., 2016) is a vehicle-egocentric image dataset collected in a computer game with pixel-wise semantic labels. It contains 24,966 images with a resolution of 1052×1914. There are 19 classes that are compatible with the Cityscapes dataset (Cordts et al., 2016). Cityscapes (Cordts et al., 2016) contains urban street images taken on a vehicle from some European cities. There are 5,000 images with pixel-wise annotations. The images have a resolution of 1024×2048 and are labeled into 19 semantic categories.

**Implementation.** We study DeepLabv2 (Chen et al., 2017) with both ResNet-50 and ResNet-101 backbone. The backbones are pre-trained on ImageNet. We also use SGD optimizer, with learning rate as $1 \times 10^{-3}$, weight decay as $5 \times 10^{-4}$, and momentum are 0.9. Batch size is set to six. We crop the images into patches of 512×512 and train the model with multi-scale augmentation (0.75 ∼ 1.25) and horizontal flipping. The model is trained for 50 epochs, and $\lambda$ for $\mathcal{L}_{\text{NCE}}$ is set as 75.

### 4.2.1 MAIN RESULTS

We also evaluate the generalization performance of our CSG on semantic segmentation. In particular, we treat the GTA5 training set as the synthetic source domain and train segmentation models on it. We then treat the Cityscapes validation sets as real target domains, where we directly evaluate the synthetically trained models. We can see that in Table 5, CSG achieves the best performance gain. IBN-Net Pan et al. (2018) improves domain generalization by carefully mix the instance and batch normalization in the backbone, while Yue et al. (2019) transfers the real image styles from ImageNet to synthetic images. However, Yue et al. (2019) requires ImageNet images during synthetic training, and also implicitly leverages ImageNet labels as auxiliary domains.

---

[3]Grad-CAM visualization method (Selvaraju et al., 2017): https://github.com/utkuozbulak/pytorch-cnn-visualizations

**Table 5:** Comparison to previous domain generalization methods for segmentation (GTA5→ Cityscapes).

| Methods | Backbone | mIoU % | mIoU ↑ % |
|---|---|---|---|
| No Adapt IBN-Net (Pan et al., 2018) | | 22.17 29.64 | 7.47 |
| No Adapt Yue et al. (Yue et al., 2019) | ResNet-50 | 32.45 37.42 | 4.97 |
| No Adapt ASG (Chen et al., 2020c) | | 25.88 29.65 | 3.77 |
| No Adapt CSG (ours) | | 25.88 35.27 | **9.39** |
| No Adapt Yue et al. (Yue et al., 2019) | | 33.56 42.53 | 8.97 |
| No Adapt ASG (Chen et al., 2020c) | ResNet-101 | 29.63 32.79 | 3.16 |
| No Adapt CSG (ours) | | 29.63 38.88 | **9.25** |

### 4.2.2 FEATURE DIVERSITY ON SEGMENTATION WITH BALANCED TRAINING SET

We further conduct visualization and quantitative measures of feature diversity on the segmentation task. Similar to section 2, we randomly sample a subset of the GTA5 training set to match the size of the Cityscapes training set. We again have similar observations: models trained on real images have relatively diverse features, and synthetic training leads to collapsed features. Here we get lower $E_s$ than classification since we follow the setting in Eq. 6 to study dense-level features. This leads to a larger total number of features on segmentation than classification.

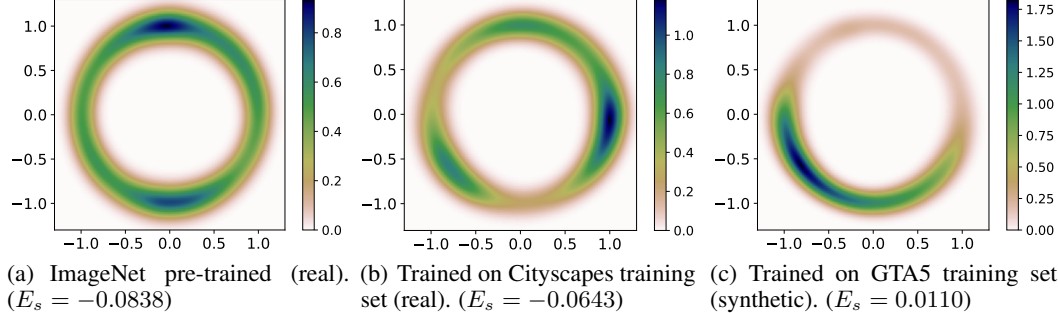

(a) ImageNet pre-trained (real). ($E_s = -0.0838$)   (b) Trained on Cityscapes training set (real). ($E_s = -0.0643$)   (c) Trained on GTA5 training set (synthetic). ($E_s = 0.0110$)

**Figure 6:** Feature diversity on Cityscapes test images in $\mathbb{R}^2$ with Gaussian kernel density estimation (KDE). Darker areas have more concentrated features. $E_s$: hyperspherical energy of features, lower the more diverse.

### 4.2.3 VISUAL RESULTS

By visualizing the segmentation results (Figure 7), we can see that as our CSG framework achieves better mIoU on unseen real images from the Cityscapes validation set, the model produces segmentation with much higher visual quality. In contrast, the baseline model suffers from much more misclassification.

## 5 RELATED WORK

**Domain generalization** considers the problem of generalizing a model to the unseen target domain without leveraging any target domain images (Muandet et al., 2013; Gan et al., 2016). The core challenge is how to close the domain gap and align feature spaces from different domains, without even seeing the target domain's data. Muandet et al. (2013) proposed to use MMD (Maximum Mean Discrepancy) to align the distributions from different source domains and train their network with

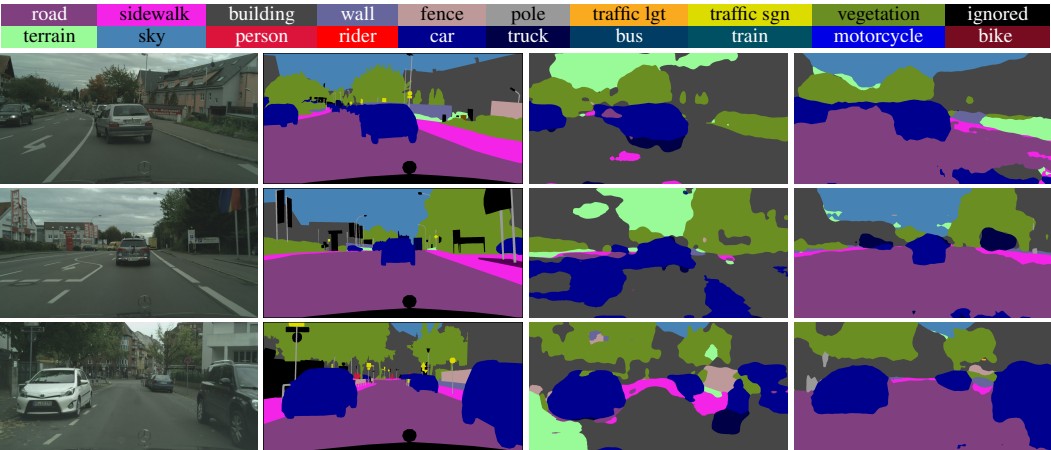

| road | sidewalk | building | wall | fence | pole | traffic lgt | traffic sgn | vegetation | ignored |
| --- | --- | --- | --- | --- | --- | --- | --- | --- | --- |
| terrain | sky | person | rider | car | truck | bus | train | motorcycle | bike |

**Figure 7:** Generalization results on GTA5 → Cityscapes. Rows correspond to sample images in Cityscapes validation set. From left to right, columns correspond to original images, ground truth, predication results of baseline (DeepLabv2-ResNet50 Chen et al. (2017)), and prediction by model trained with our CSG framework.

adversarial learning. Li et al. (2017) built separate networks for each source domain and used shared parameters for testing. By using a meta-learning approach on split training sets, Li et al. (2018) further improved generalization performance. Instance Normalization and Batch Normalization are carefully integrated into the backbone network by Pan et al. (2018) to boost network generalization. Differently, Yue et al. (2019) proposed to transfer information from the real domain as image styles to synthetic images. Most recently, (Chen et al., 2020c) formulated domain generalization as a life-long learning problem (Li & Hoiem, 2017), and try to avoid the catastrophic forgetting about the ImageNet pre-trained weights and to retain real-domain knowledge during transfer learning.

**Contrastive learning.** Noise contrastive estimation loss (Wu et al., 2018) recently becomes a predominant design choice for self-supervised contrastive representation learning (Hjelm et al., 2018; Oord et al., 2018; Hénaff et al., 2019; Tian et al., 2019; He et al., 2020; Misra & Maaten, 2020; Chen et al., 2020a). Studies show that self-supervised models can serve as powerful initializations for downstream tasks, even outperforming supervised pre-training on several. Besides engineering improvements, key factors towards better contrastive learning include employing large numbers of negative examples and designing more semantically meaningful augmentations to create different views of images. This leads to both maximize the mutual information between two views of the same instance and pushing examples from different instances apart (Tian et al., 2020b). As also observed by Wang & Isola (2020), contrastive learning tends to align the features belonging to the same instance, while scattering the normalized learned features on a hypersphere. However, most work focus on the representation learning for a real-to-real transfer learning setting where the main focus is to improve the performance of the downstream tasks. While having connections to these methods, our work pursues a different task with different motivations despite the converging techniques.

## 6 CONCLUSIONS

Motivated by the observation that models trained on synthetic images tend to generate collapsed feature representation, we make a hypothesis that the diversity of feature representation plays an important role in generalization performance. Taking this as an inductive bias, we propose a contrastive synthetic-to-real generalization framework that simultaneously regularizes the synthetically trained representations while promoting the diversity of the features to improve generalization. Experiments on VisDA-17 validate our hypothesis, showing that the diversity of features correlates with generalization performance across different models. Together with the multi-scale contrastive learning and attention-guided pooling strategy, the proposed framework outperforms previous state-of-the-arts on VisDA-17 with sizable gains, while giving competitive performance and the largest relative improvements on GTA5→Cityscapes without bells and whistles.

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
