# OpenReview forum: "Contrastive Syn-to-Real Generalization"
_ICLR.cc/2021/Conference — ICLR 2021 Poster_

### Official Review · AnonReviewer4 · 2020-10-24
**Accept**

**Rating:** 7
**Confidence:** 4

**Review:**

Synthetic-to-real generalization is an important topic of extensive practical interest. This paper motivates its work from an observation of feature diversity difference between synthetic and real training: synthetic training tends to generate less diverse or even collapsed features, whereas models trained on natural images give much diverse ones. Based on that, the author developed a contrastive “push and pull” framework to: (1) keep proximity between learned and ImageNet-pretrained features; and (2) push the feature embeddings away from each other across different images, using the feature diversity observation as the inductive bias.

Although the proposed formulation looks similar to recent contrastive representation learning, this work seems to be well motivated, and find some unique interpretations for their push and pull in the specific context of syn-to-real generalization. The authors also proposed two variants: a multi-layer contrastive loss applied to intermediate feature map groups; and a cross-task dense contrastive loss that operates on the batch level for dense supervision.

It has been known that CNN tends to focus on textures and hence learns poorly from the less texture-rich synthetic data. It is further great to see the authors show an example illustrating the collapsed representations of a synthetic model, which clearly differs from those from training on real data. This provides a feature-level clear and measurable illustration of syn-real domain gap and implies the insight what good features should behave like in syn-to-real generalization.

Experiments were on VisDA-17 (classification) and GTA5→Cityscapes (segmentation). Ablation studies were also presented on different augmentation and model components. They also proved again that the diversity of features correlates with generalization performance.

While I feel overall positive, a number of questions and concerns remain for the authors:
-	In Section 2 proof-of-concept, note that VisDA-17 training and validation sets have different sizes. Have you examined the potential confounding facor of different training steps, as longer training might be more prone to forgetting? I think a controlled experiment here could be training with two equal-sized subsets from VisDA-17 train/val.
-	Section 3.2: the authors seem to claim the two augmentations as their innovations, which may need be toned down. Using temporal averaged models inherits from MOCO, and image augmentation is now a standard in contrastive learning.
-	Further, I wonder why the authors used RandAugment, instead of using the cascaded augmentation ways adopted by MOCO-V2/SimCLR.
-	Section 3.4: I am not sure if I fully get the point of A-pool. Is that the same as adding an attention layer in the nonlinear projection head?
-	Where is the dense NCE loss applied? Was it adopted by both classification and segmentation experiments by default, or only in the segmentation? That was not specified anywhere in the paper.

---

> ### Author Response · Authors · 2020-11-22
> **Response to Reviewer 4**
>
> We appreciate your time and efforts!
>
> **More controlled experiment with equal training size**
>
> Following the suggestion, we conduct another experiment with a randomly sampled subset of VisDA-17 training set which has the same size as the validation set. We observe a similar pattern where the learned features collapse with higher energy. The results are visualized and reported in Appendix A.1.
>
> **Tone down augmentation**
>
> Many thanks. We have rewritten this part by specifically mentioning these are existing popular approaches. The rest of the descriptions are necessary for readers to correctly understand the technical details.
>
> **Why choosing RandAugment**
>
> Unlike training on natural images where the images have rich textures and enough content variations that better support the models, training on synthetic images is characterized by a lack of texture and content variations. Compared to the augmentations in SimCLR/MoCo, we found that RandAugment tends to let models better focus on structure and shape representations of the object. Our empirical study shows that augmentations from SimCLR/MoCo-v2 lead to inferior generalization performance.
>
> **Clarification on A-pool**
>
> We add A-pool before the non-linear projection head and the head takes the pooled feature as input. A-pool is motivated by our desire to focus on semantically meaningful areas to obtain a better feature representation for our contrastive framework.
>
> **Where is the dense NCE loss applied**
>
> Sorry for the confusion. The dense NCE loss is only adopted in segmentation and we have rewritten the paper to make it clearer. Note that the final loss for both classification and segmentation is denoted by Eq. 4, where $L_{NCE}$ takes the form of Eq. 5 (multi-layer and not dense) in classification while taking the form of Eq. 6 (multi-layer and dense) in segmentation.

---

> ### Comment · AnonReviewer4 · 2020-11-23
> **Thanks for the rebuttal!**
>
> The authors address all of my concerns in the rebuttal, and I decide to keep my score unchanged.

---

### Official Review · AnonReviewer2 · 2020-10-27
**Official Reivew of "Contrastive Syn-to-Real Generalization"**

**Rating:** 6
**Confidence:** 4

**Review:**

Training on synthetic data and generalizing to real test data is an important task that can be particularly beneficial for label or data-scarce scenarios. The paper aims to achieve the best zero-shot generalization on the unseen target domain real images without having access to them during synthetic training. The experiments thoughtfully demonstrate both classification and segmentation, as well as ablation studies. Visualizations and interpretations are presented in addition. The visual interpretation of feature diversity in Sec. 2 is another plus. Overall this paper is good both conceptually and experimentally. It is also well written in general.

My main concern is that the current baseline comparisons in experiment are not fully consistent nor satisfactory. As observed from Table 5, by absolute numbers (not relative margin), the proposed method is roughly aligned with ASG and IBN-Net, but lags much behind (Yue et al., 2019). That was partially explained by the authors, by saying that Yue et al. (2019) required ImageNet images during synthetic training and also implicitly leveraged ImageNet labels as auxiliary domains. However, if looking more closely, even the baseline ResNet-50 mIoU sees a big gap between Yue et al. (2019), and the proposed method as well as the two others. It is unclear and unconvincing to me why the same baseline can perform so differently among those methods, and I think this might potentially undermine the experiment reliability/reproducibility and deserves more clarification from the authors.

One more nitpick is that this paper shows no figure in experiments. Given there is some extra space, the authors may want to visualize some classification and segmentation results, displaying both success and failure cases.

Typos:

a novel framework that leverage  -> should be “leverages”

the diversity of learned feature embedding play -> should be “plays”

---

> ### Author Response · Authors · 2020-11-22
> **Response to Reviewer 2**
>
> We appreciate your time and efforts!
>
> **Comparison to Yue et al. (2019)**
>
> We fully understand the reviewers’ concerns and share the same dissatisfaction with the reviewers. We have tried hard to ensure our setting is fair and the results are solid. To this end, we have spent efforts by trying the items listed below:
>
> 1) Different learning rates
>
> 2) Different input/output resolutions
>
> 3) Different header designs (FCN/DeepLabv2)
>
> In summary, our synthetic training on GTA5 is highly aligned with a number of previous literature including IBN-Net (Pan et al. 2018), ASG (Chen et al. 2020), as well as other works from this community such as FCAN (29.2% with ResNet-101) [1], DCAN (29.8% with ResNet-101) [2] and ADR (25.3 with ResNet-50) [3]. Yue et al. 2019 is a great work but we fail to get sufficient details to reproduce their reported GTA5 baseline after spending many efforts, including email-querying the authors. We conjecture that their quite high baseline number might demand unusually good (but non-standard) engineering implementations to achieve. If that is the case, we opt to stand with the remaining community's baseline numbers which are consistent and more easily reproducible.
>
> It is also worth mentioning that any improvement in naive synthetic training in general transfers very well to improvement in CSG training. As a result, obtaining a higher baseline result is by nature aligned with our ultimate goal. There are indeed many ways to improve the testing performance, but are not common recipes. Our general experience is that synthetic training on GTA5 can be brittle and unstable. It is often difficult to keep the performance always higher than 30% without bells and whistles such as training with extremely small numbers of steps, hand designing layer-wise learning rates, and picking stopping points. These bells and whistles are not generalizable across datasets and require reference to validation accuracies, which implicitly defeats the zero-shot generalization purpose of this paper. Our main purpose is not to chase absolute state-of-the-art with these bells and whistles, but to best represent the general synthetic training experience people encounter and look for solid improvements to such experience.
>
> [1] Zhang et al., Fully convolutional adaptation networks for semantic segmentation, CVPR 2018.
>
> [2] Wu et al., DCAN: Dual channel-wise alignment networks for unsupervised scene adaptation, ECCV 2018.
>
> [3] Saito et al., Adversarial dropout regularization, ICLR 2018.
>
>
> **Visualize classification and segmentation results**
>
> Following your great suggestion, we added visualization and quantitative analysis of feature diversity for segmentation in Sec. A.2. In summary, we observe similar phenomenons of collapsed features in segmentation with naive synthetic training, whereas training on real images promotes feature diversity. We’ve also added additional visualizations in Appendix Sec. B.
>
> **Typos**
>
> Many thanks. We have fixed the typos.

---

### Official Review · AnonReviewer1 · 2020-10-28
**The observed phenomenon is interesting but the proposed method is not a well-motivated solution.**

**Rating:** 6
**Confidence:** 4

**Review:**

Summary

This paper focuses on the domain generalization problem where the source domain contains synthetic data. An interesting phenomenon is observed in this paper: the diversity of the learned feature embeddings plays an important role in the generalization performance. Then, this paper presents a method to address the syn-to-real generalization problem by combining augmentation, contrastive loss and attention pooling techniques. In general, the observed phenomenon is interesting but the proposed method is not a well-motivated solution. Besides, the researched problem is not a general one, which limits the impact of this paper.

Pros:

1. When the source domain only contains synthetic data, this paper observes that the diversity of the learned feature embeddings plays an important role in the generalization performance in Section 2. From Figure 2, it is clear that the synthetic data are very different from real data in the view of feature diversity. This phenomenon is interesting and maybe motivate more works regarding syn-to-real problems (not only the domain generalization problem).

Cons:

1. This paper provides a new view of addressing the syn-to-real generalization problem. However, the solution presented in this paper is not novel and is not connected with the new view very well. It is not novel enough by only combining augmentation, contrastive loss and attention pooling techniques.

2. The experiments are not enough. For general interest, this paper should also present results when using the proposed method to address ordinary domain generalization problems. The researched problem is not a general one, which limits the impact of this paper.

3. There are many typos in this paper (even in the abstract: "that leverage" should be "that leverages").

4. It is unclear why we should use augmentation and A-pool. The motivations behind the two techniques are unclear. Why can they improve the accuracy of the proposed method?

---

> ### Author Response · Authors · 2020-11-22
> **Response to Reviewer 1**
>
> We appreciate your time and efforts!
>
> **Presented solution not novel and not well connected with the new view**
>
> Our proposed methods are strongly motivated to serve the main observation: since synthetic training leads to collapsed embeddings, it is important to leverage diversity as an inductive bias to improve the synthetic training quality. It is this observation that leads to our proposed “push-and-pull” strategy, where “push” promotes features diversity and “pull” encourages knowledge distillation. This “push-and-pull” strategy further motivated the contrastive learning formulation, augmentation and A-pool. Augmentation (both image and model level) is a necessary module rather than an add-on for good reasons: Besides conventionally being a highly coupled part of contrastive learning, it creates diverse views that help to overcome overfitting in synthetic training and forces the model to focus on semantically more meaningful parts/structures. In addition, A-pool emphasizes object-centric attention which guides the contrastive learning to semantically more meaningful areas and is uniquely motivated. As also noted by other reviewers, our idea is novel and the proposed approaches are well motivated.
>
> **Present results on ordinary domain generalization**
>
> We fully understand why you have this concern but we kindly disagree. Both synthetic-to-real generalization and domain generalization on natural images present valid and important research topics. There have been many research works on synthetic-to-real generalization/adaptation, and the topic is significant in general robotics/visual reinforcement learning/simulation. The two problems have different motivations despite sharing some overlap, and there is no superiority of one over another per se. We notice that the importance of synthetic-to-real is also acknowledged by some other reviewers as having extensive practical interest.
>
> Our idea and proposed methods are uniquely motivated by our observations in synthetic training, and are tailored to specifically address this problem, even though similar techniques may turn out to benefit domain generalization on natural images as well. Our observation that synthetic training tends to result in collapsed features, together with the visualization and quantification using hyperspherical energy, present unique contributions to the community. We believe that the current design of experiments in this work is adequate to support our contributions. We also believe that machine learning research isn’t just about universality, but is also about considering data distributions, priors and inductive biases.
>
> **Typos**
>
> Many thanks. We have fixed the typos.

---

> > ### Comment · AnonReviewer1 · 2020-11-23
> > **My concerns still exist.**
> >
> > Thanks for replying to my previous comments. However, I still find that my previous concerns still exist. the contrastive learning formulation, augmentation and A-pool are just combined to become the proposed method, which does not make me feel the strong connection between the proposed method and the interesting observed phenomenons. Besides, results on ordinary domain generalization should be presented in the appendix, which can make readers know the performance of the proposed method in general scenarios.
> >
> > I am disappointed that my previous comments are ignored in the revision. However, considering the observed phenomenons are interesting, I still keep my previous score. I hope that my concerns will be addressed in the final version (if accepted).

---

> > > ### Author Response · Authors · 2020-11-25
> > > **Thank you very much for your suggestions and opinions!**
> > >
> > > We will continue working on experiments of generalization problems in more scenarios, and strive to report the results upon acceptance.

---

### Official Review · AnonReviewer3 · 2020-10-29
**idea is novel, more technical details needed**

**Rating:** 6
**Confidence:** 4

**Review:**

This paper was motivated from an observation the common lack of texture and shape variations on synthetic images often leads the trained models to learning only collapsed and trivial representations without any diversity. The authors made a hypothesis that the diversity of feature representation would pay an important role in generalization performance and can be taken as an inductive bias.

Seeing that, they proposed a synthetic-to-real generalization framework that simultaneously regularizes the synthetically trained representations while promoting the diversity of the features to improve generalization. Their strategy can be exactly formulated with a contrastive loss, which reminds me of Wang & Isola (2020) but was also customized for the synthetic-to-real generalization scenario. The framework was further enhanced by the multi-scale contrastive learning and an attention-guided pooling strategy. Besides, the dense contrastive loss (6) provided spatially denser patch-level supervision; that may be a novel idea that I haven’t seen before. However, the authors did not clarify where and how they use loss in their experiments.

Experiments on VisDA-17 and GTA5 supported the hypothesis: though assisted with ImageNet initialization, fine-tuning on synthetic images tends to give collapsed features with poor diversity in sharp contrast to training with real images. This indicates that the diversity of learned representation could play an important role in synthetic-to-real generalization. Their experiments showed that the proposed framework can improve generalization by leveraging this inductive bias and can outperform previous state-of-the-arts without bells and whistles.

I also feel more analysis and insights could have been provided for the segmentation experiments in 4.2. Currently there is no more information beyond Table 5. For example, some feature diversity measure like Table 1 could be reported for segmentation too, since revealing the feature diversity inductive bias is the main novelty in this paper. Also, more clarifications are needed on comparing the settings fairly with prior work like Pan et al. (2018) and Yue et al. (2019).

---

> ### Author Response · Authors · 2020-11-22
> **Response to Reviewer 3**
>
> We appreciate your time and efforts!
>
> **Where and how losses are used in experiments**
>
> Sorry for the confusion. The dense NCE loss is only adopted in segmentation and we have rewritten the paper to make it clearer. Note that the final loss for both classification and segmentation is denoted by Eq. 4, where $L_{NCE}$ takes the form of Eq. 5 (multi-layer but not dense) in classification, and takes the form of both Eq. 6 (multi-layer and dense) in segmentation. More specifically, $L_{Task}$ is a supervised cross-entropy loss calculated with ground truth, and $L_{NCE}$ is for representation learning on synthetic images without labels.
>
> **More analysis and insights on segmentation**
>
> We followed your suggestion and conducted this additional analysis and visualizations. In summary, on segmentation, we make similar observations where synthetic training leads to collapsed features. Our quantitative analysis on feature diversity also indicates that features from the ImageNet initialized model and the CSG trained model have lower hyperspherical energy.
> The results are reported in Appendix Sec. A.2 and B.2. Specifically, we train on both synthetic data (GTA5 training set) and real data (Cityscapes training set) where we unify the training set size and training configuration following R4’s suggestion.
>
> **Clarification on comparing fairly with Pan et al. (2018) and Yue et al. (2019)**
>
> We tried to ensure our setting is fair and the results are solid. To this end, we spent the following efforts:
>
> 1) Different learning rates
>
> 2) Different input/output resolutions
>
> 3) Different header designs (FCN/DeepLabv2)
>
> In summary, our synthetic training on GTA5 is highly aligned with a number of previous literature including IBN-Net (Pan et al. 2018), ASG (Chen et al. 2020), as well as other works from this community such as FCAN (29.2% with ResNet-101) [1], DCAN (29.8% with ResNet-101) [2] and ADR (25.3 with ResNet-50) [3]. Yue et al. 2019 is also a great work but we fail to get sufficient details to reproduce their reported GTA5 baseline after spending many efforts, including email-querying the authors. We conjecture that their quite high baseline number might demand unusually good (but non-standard) engineering implementations to achieve. If that is the case, we opt to stand with the remaining community's baseline numbers which are consistent and more easily reproducible.
>
> [1] Zhang et al., Fully convolutional adaptation networks for semantic segmentation, CVPR 2018.
>
> [2] Wu et al., DCAN: Dual channel-wise alignment networks for unsupervised scene adaptation, ECCV 2018.
>
> [3] Saito et al., Adversarial dropout regularization, ICLR 2018.

---

### Author Response · Authors · 2020-11-22
**Response to All Reviewers**

We sincerely thank all the reviewers for constructive and insightful comments. We summarize our changes here and address specific questions from individual reviewers separately below.

**Summary of changes**
* Added additional visualization and quantitative analysis for feature diversity on VisDA-17 to address better-controlled experiment in Appendix Sec A.1 (R4)
* Added additional visualization and quantitative analysis for feature diversity on GTA5 -> Cityscapes in Appendix Sec A.2 (R2/R3)
* Added GradCam visualization on VisDA-17 in Appendix Sec B.1 (R2)
* Added visualization for semantic segmentation in Appendix Sec B.2 (R2/R3)
* Fixed typos and rewrote part of the paper (R1/R2/R4)

**Reproducibility**: We will release the code upon acceptance.

---

### Decision · Program_Chairs · 2021-01-07
**Final Decision**

**Decision:**

Accept (Poster)

**Comment:**

The paper raised a natural question: why good synthetic images can be not so good at training/fine-tuning models for downstream tasks (e.g., classification and segmentation)? This problem is named synthetic-to-real (domain) generalization (where syn/real images are regarded as from the source/target domain), and it is of practical importance when using GAN-like methods given limited real images for training. The authors found that the answer to the question is the diversity of the learned feature embeddings, and argued/advocated that we should encourage such diversity when training on syn images in order to better approximate training on real images. To this end, a novel contrastive synthetic-to-real generalization framework was proposed and shown effective in the well designed experiments.

Overall, the quality is above the bar. While some reviewers had some concerns about the applicability and the motivations for the algorithm design, the authors have done a particularly good job in the rebuttal. After the rebuttal, we all think the paper should be accepted for publication.

I have some comments on the writing. The introduction claiming so many things has only 4 citations, especially the first two paragraphs have no citation. While I do think what claimed there are correct, the authors should include certain supportive evidences after each claim by themselves. Moreover, while I do think the problem hunting part is well motivated, the problem solving part needs its own motivation/justification. When two or more components are combined in a proposal, why this component is chosen and is there other choice that can achieve the same purpose (this concern has also been raised by reviewers)? I believe the components are not randomly chosen among possible candidates (e.g., "we further enhance the CSG framework with attentional pooling"), but for writing a paper, the authors should explain the motivation for the algorithm design because we cannot know the motivation unless they tell us.